# Effects of First-Time Experiences and Self-Regulation on College Students' Online Learning Motivation: Based on a National Survey during COVID-19

**Gege Li [1], Heng Luo [1,\*] , Jing Lei [2,\*], Shuxian Xu [1] and Tianjiao Chen [1]**

[1] Faculty of Artificial Intelligence in Education, Central China Normal University, Wuhan 430079, China; ligg323@mails.ccnu.edu.cn (G.L.); xushuxian@mails.ccnu.edu.cn (S.X.); chentianjiao@mails.ccnu.edu.cn (T.C.)

[2] Department of Instructional Design, Development and Evaluation, Syracuse University, 804 University Ave., Syracuse, NY 13244, USA

\* Correspondence: luoheng@mail.ccnu.edu.cn (H.L.); jlei@syr.edu (J.L.)

**Abstract:** The COVID-19 pandemic has forced many college students in developing countries to engage in online learning for the first time, and the sudden transit has raised concerns regarding students' competencies for, perception of, and attitude towards online learning. To address those concerns, this study measured three essential constructs of online learning (self-regulated learning, perceived presences, and learning motivation) based on a national survey in China (N = 12,826) and employed structural equation modeling to investigate their intertwined relationship. The study results reveal that (1) college students' academic achievement cannot effectively predict their self-regulated learning in an online learning context; (2) self-regulation can be further differentiated into general and task-specific strategies with a varying impact on three types of presences; (3) online learning motivation is best predicted by cognitive presence, followed by social presence and teaching presence; and (4) the path of task-specific self-regulated learning → cognitive presence → online learning motivation generates the largest positive compound effect. Implications for online teaching and learning practice are also discussed through the stakeholder perspectives of students, teachers, and platform developers.

**Keywords:** online learning motivation; self-regulated learning; community of inquiry; structural equation modeling; survey research; China; higher education; COVID-19

## 1. Introduction

While online learning has experienced a steady increase in practice for the past few decades [1,2], it became a mandated educational norm with worldwide impact until more recently, in the wake of massive school closures due to COVID-19. Many students, especially those in the developing countries, engaged in online learning for the first time [3]. With its enhanced accessibility and flexibility, online learning ensures the continuity of education during the pandemic, and provides valuable lessons for designing more inclusive, customized, and student-centered learning experiences in the post-pandemic world [4,5]. Similar to the learning in physical classrooms, the success of online learning is largely determined by three pertinent constructs, learner motivation, educational experience, and self-regulation [6–8], given that the basic infrastructure and technical resources for online learning are met. To extend our understanding of online learning for improved quality and equity, the three constructs and their interrelations merit our special attention.

Motivation is a central tenet of teaching and learning [9], which effectively affects students' learning performance [6], engagement [10], and attitude [11]. It is especially important for the online learning environment due to the proven association between learner motivation, satisfaction, and course retention [12–15]. Moreover, understanding motivation towards online learning is of particular significance in the post-pandemic era,

as the forced shift to online education can further diminish the lines between physical and virtual learning spaces [16]. Interestingly, learner motivation is often considered as a predictor rather than an outcome of online learning in the literature [17–19], even though it is shaped retrospectively by the educational experience perceived by online learners, especially for first-time online learners.

To capture the complexity of online learning as a transactional educational experience, Garrison et al. (2001) devised the Community of Inquiry (CoI) framework, featuring three interrelated conceptual components that are fundamental to meaningful learning in an online context, known as teaching presence, social presence, and cognitive presence [20]. The composition of the presences has been validated across populations and disciplines [21–23]. Although the original intent of CoI was to describe rather than predict [24], recent studies have attributed positive learning outcomes to the multifaceted online learning experience dictated by CoI [25–27]. However, the causal relationships between CoI and learner motivation are rarely investigated in the literature, despite the proven importance of motivational beliefs for adopting educational innovations [28,29].

While the CoI framework describes the social–cognitive nature of online learning at a macro level [30,31], it fails to address the individual-level differences in self-regulated learning (SRL) [27,32]. SRL emphasizes learners' proactive control and regulation of their own learning processes [11,33], which is particularly essential for the online learning environment, featured by greater learner autonomy and lesser instructor presence [34,35]. SRL is found to significantly influence student's perception of CoI-related experiences, making it a potential fourth component in the CoI framework as a learning presence [32,36]. The correlational effects between SRL and motivation were also reported in the literature, with successful SRL associated with positive motivation [11,35,37]. Yet, a comprehensive investigation of SRL in light of CoI and motivation is still lacking.

Although students' perception and self-regulation of their online learning process plays a crucial role in predicting online learning motivation, few studies have examined the reciprocal relationships among those multifaceted constructs comprehensively through the theoretical lens of both CoI and SRL. There is also a need for more contemporary findings about online learning due to the enormous paradigm changes induced by recent technology advancement (e.g., synchronous presence, live streaming communication, and multi-device access). Additionally, we found that most of the related studies were based on survey data from a single institute with a relatively small sample size [23], and factors essential for online SRL such as academic achievement and disciplines were often excluded from analysis. Those limitations undermine the external and internal validity of the research findings.

To address these research limitations, we conducted a large-scale survey (N = 12,826) in China during COVID-19 to assess college students' online learning experience and attitude. Based on the survey results, the present study aims to further our understanding of the interplay among CoI, SRL, and online learning motivation, while taking into consideration various demographic and contextual factors.

## 2. Relevant Literature

### 2.1. Definition of Key Concepts

#### 2.1.1. Learning Motivation

Brophy (2010) defines motivation as "a theoretical construct to explain the initiation, direction, intensity, persistence, and quality of behavior, especially goal-directed behavior" (p. 3) [38]. Researchers have classified learning motivation into intrinsic and extrinsic motivation [39]. Intrinsic motivation is concerned with the activity itself: a motivating learning activity should be enjoyable or optimally challenging, allowing learners to initiate their learning behaviors that reflect on their own values and hobbies [40]. In contrast, extrinsic motivation is generally sustained by rewards or other external incentives [41]. Compared to brick-and-mortar classrooms where a variety of reward and punishment are administered by the onsite instructor, the online learning environment requires learners to be more intrinsically motivated due to its greater demand for autonomy, competence,

and self-discipline [42,43]. Consequently, learning motivation investigated in this study refers to intrinsic motivation. According to Lin et al. (2020), intrinsic motivation is a multifaceted construct comprising four dimensions: interest, competence, value, and pressure [40].

### 2.1.2. CoI and the Interplay of Three Presences

CoI is a conceptual framework initially developed by Garrison et al. (2000) to describe effective higher education experiences [44]. It was later used to clarify the behaviors and processes required for effective knowledge construction in online learning environments by describing three key elements: teaching, social, and cognitive presence [20]. Teaching presence refers to the design and organization of teaching process perceived by learners [45]. Social presence refers to the ability of participants to express themselves through online learning platforms in social and emotional ways with teachers, peers, content, and resources [46]. Cognitive presence is regarded as the "focus and success of the learning experience" [47], as it reflects online learners' construction and verification of meaning through critical communication and higher-order thinking [48].

The inter-relationships among the three presences in CoI have been investigated thoroughly, with several major findings reported in the literature. First, teaching presence proved to be a predictor of both social and cognitive presences [49,50]. Well-designed tasks, facilitated discourse, and pedagogical activities are core features of teaching presence that move students through phases of collaborative inquiry [51]. Second, cognitive presence is often considered as the outcome of both teaching and social presences [52,53]. Particularly, direct instruction and group cohesion are speculated to induce higher levels of cognitive presence [51]. Lastly, statistical results from various structural equation models reveal that teaching and cognitive presence can have a direct or indirect relationship, and social presence acts as an intermediary construct [31,54]. Synthesizing the literature findings, we propose the following hypotheses regarding the interplay of three CoI presences:

**Hypothesis 1 (H1).** *Positive correlation exists among the three CoI presences.*

**Hypothesis 1a (H1a).** *Teaching presence has a positive effect on social presence.*

**Hypothesis 1b (H1b).** *Teaching presence has a positive effect on cognitive presence.*

**Hypothesis 1c (H1c).** *Social presence has a positive effect on cognitive presence.*

### 2.1.3. SRL and Its Classification

As defined by Zimmerman and Schunk (2011), SRL is a dynamic process where learners continuously adjust their learning to attain personal goals [55]. According to Barnard et al. (2008), SRL consists of six key strategies: goal setting, environment structuring, task strategy, time management, help seeking, and self-evaluation [56]. We believe all those SRL strategies apply to the online learning process, except for environment structuring, as online courses are often highly structured, allowing little leeway for environment construction. Moreover, Gagné et al. (2005) argued that learning strategies can be classified into task-specific and general based on the nature of competence [57]. Consequently, SRL investigated in this study featured five SRL strategies for online learning and divided into two categories: task-specific strategies and general strategies.

The task-specific strategies category includes strategies of goal setting, task strategy, and time management, as different learning tasks require students to select their learning goals and task-strategies, and often lead to adjusted time on task. The general strategies category involves help seeking and self-evaluation, because they are habitual tendencies regardless of learning tasks and contexts. Help seeking is a proactive social learning behavior that indicates a high level of agency, which is essential to complete online learning tasks [58], and self-evaluation informs the individual decision of learning goals, strategies,

and time [59]. In other words, we believe general SRL can effectively predict task-specific SRL, and hypothesize the following:

**Hypothesis 2 (H2).** *General SRL has a positive effect on task-specific SRL.*

*2.2. Relationships among the Key Concepts*

2.2.1. Learning Motivation and Community of Inquiry

Learning motivation was found to profoundly influence students' autonomous learning experience [60,61], particularly the development and maintenance of community awareness during the learning process [6]. Naturally, the relationship between learning motivation and CoI experiences was examined in the literature, which revealed a positive correlation between learning motivation and the CoI presences [19,62]. For instance, Ferrer et al. (2020) contributed the improvement of online learning motivation to teachers' competences to facilitate discourse and learning activities in online settings [6]. In addition to teacher support, Whip and Chiarelli (2004) also identified peer support as a key motivational influence on the completion of an online course [63]. Furthermore, Hartnett (2012) proved that the connection between students' cognitive participation and motivational quality exists in online settings just like in traditional classrooms [64]. The above findings regarding the relationship between motivation and CoI-presences were validated by Kim (2015) with the statistical results from multiple linear regression and path analysis [65]. Consequently, we propose the following hypotheses:

**Hypothesis 3 (H3).** *CoI presences can predict online learning motivation.*

**Hypothesis 3a (H3a).** *Teaching presence has a positive effect on online learning motivation.*

**Hypothesis 3b (H3b).** *Social presence has a positive effect on online learning motivation.*

**Hypothesis 3c (H3c).** *Cognitive presence has a positive effect on online learning motivation.*

2.2.2. Self-Regulated Learning and Community of Inquiry

The three presences in CoI are considered insufficient to describe the metacognitive aspects of online learning, with key concepts such as self-efficacy, self-regulation, and student agency not accounted for [66,67]. As a result, researchers have proposed a fourth element, named learning presence, to the CoI framework [68,69]. The essence of learning presence is SRL; according to Zimmerman (2008), "it was viewed as especially important during personally directed forms of learning, such as discovery learning, self-selected reading, or seeking information from electronic source" (p. 167) [70]. The electronic and self-directed nature of online courses highlights the key role of SRL in online learning, and indicates the need to examine its influence on CoI-related experiences.

The literature in general regards SRL as an independent variable that effectively predicts CoI presences [36,62]. For instance, Selcan and Zahide (2018) conducted a linear regression to identify SRL as a significant predictor of the three CoI presences [62]. Shea and Bidierano (2010) further revealed self-efficacy and self-regulation as partial mediators of the interplay among the three presences based on the path analysis results [69]. However, most studies treated SRL as a single construct when investigating its influence on CoI-presences, ignoring the distinction between task-specific and general SRL strategies. To address such limitations, we propose the following hypotheses:

**Hypothesis 4 (H4).** *Online students' SRL positively predicts their CoI-presences.*

**Hypothesis 4a (H4a).** *General SRL has a positive effect on teaching, social, and cognitive presence.*

**Hypothesis 4b (H4b).** *Task-specific SRL has a positive effect on teaching, social, and cognitive presence.*

### 2.2.3. Demographic Variables and Self-Regulated Learning

A variety of demographic variables were found to influence online students' SRL, including seniority [71], sex [72], grade level [73], education level [74], academic achievement [75,76], and discipline [73]. In general, the literature suggests that students who are older, female, well-educated, and high achieving tend to have better SRL skills, and domains of learning is a potential moderator of the relationship between SRL and learning outcomes. In addition to the aforementioned demographic variables, we are particularly interested in the role of college type on students' SRL: Due to the current exam-based college admission system in China, the prestigious status of a college is a key indicator of its students' learning aptitude, motivation, and metacognitive skills. Consequently, we propose the following hypotheses regarding the demographic variables that influence online students' SRL:

**Hypothesis 5 (H5).** *Demographic factors affect online students' SRL.*

**Hypothesis 5a (H5a).** *Sex has a positive effect on general and special SRL.*

**Hypothesis 5b (H5b).** *Academic rank has a positive effect on general and special SRL.*

**Hypothesis 5c (H5c).** *Field of program has a positive effect on general and special SRL.*

**Hypothesis 5d (H5d).** *Seniority has a positive effect on general and special SRL.*

**Hypothesis 5e (H5e).** *Type of college has a positive effect on general and special SRL.*

Therefore, a conceptual model composed of above hypotheses was constructed, as illustrated in Figure 1.

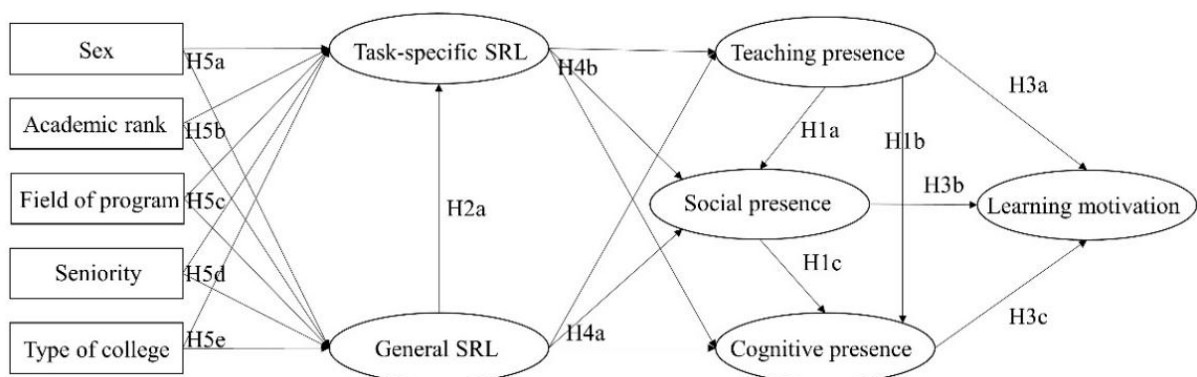

**Figure 1.** The hypothesized research model.

### 3. Method

*3.1. Research Context*

The outbreak of COVID-19 in early 2020 caused severe postponement or complete suspension of the spring semester for universities and colleges in China, forcing hundreds of millions of college students to study online for an extended period (3–5 months). The unprecedented educational experience provided a unique opportunity to investigate college students' SRL, CoI-related experiences, and motivation in the online learning environment, and provided the research context for the current study. To collect students' perception of and attitude towards their online learning experience, we conducted a national survey between 19 June and 10 July 2020, shortly after most universities and colleges ended their teaching schedule. The survey questionnaire was distributed electronically to a diverse population of college students nationwide in different provinces by means of college teacher reference and social media promotion.

A total of 28,641 questionnaires were retrieved initially, and measures were taken to screen and remove invalid ones (n = 5795) based on the following three criteria: (1) Unreasonable completion time, completion time within 60 s was deemed as too short for answering the questionnaire, casting doubts regarding the legitimacy of data. (2) Identical ratings, students giving identical ratings throughout the questionnaire indicates a lack of accuracy and careful consideration. (3) Missing data, student responses with over 50% null answers were considered as void questionnaires due to a lack of effective information. Among the remaining 22,846 valid questionnaires, our further analysis revealed that submissions from third tier and vocationally colleges were disproportionally larger in comparison with the current college composition structure in China. To boost the representativeness of survey data, we used the random trimming approach to reduce the sample size of third tier and vocational college students to fit the population distribution. Eventually, we selected a total of 12,826 collected questionnaires for the ensuing structural equation modelling analysis.

*3.2. Participants*

The questionnaire was distributed to nine provinces in China, with a large proportion in Hubei. Participants were mainly from Hubei province for two reasons. First, the COVID-19 pandemic was initially reported in Wuhan city, Hubei province, which required government taking immediate response. Hubei suffered seriously from the pandemic, which lasted the longest time, and had a complete semester of online learning. Second, the number of colleges in Hubei province ranks third in China, and have a slightly above average level in education. Therefore, we considered Hubei a representative province of online learning in China. We also included other eight provinces to further boost the representativeness of the survey sample.

As shown in Table 1, the participants were generally equally distributed in sex. Most of the participants were under the age of 21 and were within their first three years of undergraduate program, with few aged 25 and above. In terms of college rank, the number of participants decreased as the colleges became more selective. Such pyramid-shaped distribution mirrors the college composition structure in China. The fields of program were roughly equally distributed, including social science (e.g., arts, history, and education), basic science (e.g., science, engineering, and medicine), and mixed (e.g., economics and military). We also identified the class rank among the participants based on their self-evaluation. Among them, nearly half of students ranked medium, while with a few students ranked low.

*3.3. Instruments*

The questionnaire consisted of two sections, with a total of 56 items. The first section included six items collecting students' demographic information, as listed in Table 1. The second section comprised 50 items that had students evaluate their online learning experiences, including self-regulated learning, CoI presences, and learning motivation. All items in the second section were measured using a five-point Likert scale, from strongly disagree (1) to strongly agree (5). The complete questionnaire is listed in Appendix A.

**Table 1.** Demographic information of the surveyed participants.

| Demographics | Level | Sample Size | Percentage | Total |
|---|---|---|---|---|
| Sex | Male | 5230 | 40.8% | 12,826 |
| | Female | 7596 | 59.2% | |
| Age | 19 or below | 5620 | 43.8% | 12,826 |
| | 20–21 | 5887 | 45.9% | |
| | 22–24 | 1221 | 9.5% | |
| | 25 and above | 98 | 0.8% | |
| Seniority | Year 1–2 Undergraduate | 6303 | 49.1% | 12,826 |
| | Year 3 Undergraduate | 6091 | 47.5% | |
| | Year 4+ Undergraduate | 183 | 1.4% | |
| | graduate student | 249 | 1.9% | |
| College rank | Top Tier | 1251 | 9.8% | 12,826 |
| | Second Tier | 2300 | 17.9% | |
| | Third Tier | 3827 | 29.8% | |
| | Vocational College | 5448 | 42.5% | |
| Field of program | Social Science | 3655 | 28.5% | 12,826 |
| | Basic Science | 4916 | 38.3% | |
| | Mixed | 4255 | 33.2% | |
| Academic rank | Top rank | 1091 | 8.5% | 12,826 |
| | Above average | 3965 | 30.9% | |
| | Medium | 5675 | 44.2% | |
| | Below average | 1512 | 11.8% | |
| | Ranking low | 583 | 4.5% | |

### 3.3.1. Self-Regulated Learning (SRL)

A total of 13 items were adopted from the questionnaire developed by Barnard and Lan (2008), which measures students' SRL in five aspects: goal setting, task strategy, time management, help seeking, and self-evaluation [56]. Sample items included, "I took detailed notes when learning online" and "I know that online learning is time consuming, so I allocate extra time for it".

### 3.3.2. CoI Framework

The measurement of CoI framework is based on the questionnaire developed by Arbaugh et al. (2008), which consists of 24 items measuring students' perceptions towards online teaching presence, social presence, and cognitive presence [22]. Sample items included, "The teacher clearly communicated important course goals" (teaching presence), "Some of my classmates in the online courses impressed me a lot" (social presence), and "The learning tasks in the course increased my interests" (cognitive presence).

### 3.3.3. Learning Motivation (LM)

LM was measured with 13 items adopted from the questionnaire developed by Lin et al. (2020), containing four categories: interest, competence, value, and pressure [40]. Sample items included "I think the learning tasks in online learning are very interesting" (interest) and "I am competent for the learning tasks in online learning" (competence).

### 3.4. Data Analysis

We conducted two major approaches for the data analysis. The first approach was descriptive and correlational analysis. The descriptive analysis was performed to estimate students' descriptive statistics among different constructs. The correlational analysis was performed to explore the relationships among the measurement variables. The second approach was structural equation modelling (SEM) analysis to determine the validity of the hypotheses. The following four statistical indices were reported to assess the fitness of the model: Chi-square, CFI, IFI, and RMSEA. A non-significant Chi-square indicates a

good fit of the model. The values for CFI and IFI greater than 0.9 indicated an acceptable fit [77]; while an RMSEA lower than 0.08 indicated an adequate fit. The descriptive and correlational analysis was performed using IBM® SPSS® software platform (version 21), and the SEM analysis was performed in IBM® SPSS® Amos software (version 21).

## 4. Results

*4.1. Descriptive and Correlational Statistics*

Table 2 demonstrates the means, standard deviations, and correlations between the key constructs. According to Hu and Bentler (1999), the basic statistical assumptions for structural equation modelling analysis were met in such a data set [78]. The mean ratings for the three CoI presences, SRL, and LM were all above the neutral point (i.e., 3 in the five-point Likert scale), ranging from 3.69 to 3.96, and those constructs were moderately correlated with each other. Contrarily, college students' demographic factors seemed to have limited influence on their online learning experiences. Compared to sex and seniority, the factors of college rank and academic rank reported a stronger correlation with SRL and LM.

**Table 2.** Means, standard deviations, and correlations of the key constructs.

|  | Mean | SD | 1 | 2 | 3 | 4 | 5 | 6 | 7 | 8 | 9 |
|---|---|---|---|---|---|---|---|---|---|---|---|
| 1 Sex | - | - | 1 | | | | | | | | |
| 2 Senority | - | - | −0.06 | 1 | | | | | | | |
| 3 College rank | - | - | −0.20 | 0.20 | 1 | | | | | | |
| 4 Academic rank | - | - | −0.01 | −0.07 | 0.01 | 1 | | | | | |
| 5 TP | 3.96 | 0.74 | 0.02 | −0.03 | −0.08 | −0.05 | 1 | | | | |
| 6 SP | 3.87 | 0.75 | 0.03 | −0.02 | −0.09 | −0.06 | 0.85 | 1 | | | |
| 7 CP | 3.89 | 0.76 | 0.02 | −0.01 | −0.08 | −0.09 | 0.84 | 0.89 | 1 | | |
| 8 SRL | 3.79 | 0.76 | 0.03 | −0.03 | −0.13 | −0.14 | 0.74 | 0.78 | 0.84 | 1 | |
| 9 LM | 3.69 | 0.71 | 0.03 | −0.02 | −0.15 | −0.10 | 0.70 | 0.77 | 0.80 | 0.86 | 1 |

Note: SD, standard deviation; TP, teaching presence; SP, social presence; CP, cognitive presence; SRL, self-regulated learning; LM, learning motivation.

*4.2. Measurement Model*

To ensure the good reliability and validity of the questionnaire, preliminary tests were conducted to check the unidimensionality of the items in three main constructs (presence, SRL, and LM). According to Sanchez (2013), an item with a factor loading greater than 0.7 indicates it can represent most of the latent construct [79]. Based on such criteria, we removed two items whose factor loading values were lower than 0.7 (0.298 and 0.436) from the measurement of learning motivation.

Table 3 presents the reliability and validity about the summary of items. The reliability of all construct measurement was calculated used Cronbach's α. According to Nunnally (1978), the α value above 0.7 indicates a high reliability of measurement [80]. In this study, most α values were greater than 0.7, suggesting an acceptable reliability for the overall questionnaire as well as its sub-sections.

The validity of constructs was determined by convergent and discriminant validity. The acceptable convergent validity required factor loading values greater than 0.7, composite reliability (CR) values greater than 0.6, and average variance extracted (AVE) values greater than 0.5 [81]. As seen in Table 3, such cut-off criteria were met in the questionnaire, indicating a good convergent validity of the instrument. Additionally, the criteria for acceptable discriminant validity required the square roots of the AVE ($\sqrt{\text{AVE}}$) to be greater than the correlation coefficients with other constructs [82]. In this study, the $\sqrt{\text{AVE}}$ value of the constructs ranged from 0.729 to 0.924, greater than the correlation coefficients, indicating an acceptable discriminant validity of the questionnaire as well.

**Table 3.** Statistical summary of constructs, reliability, and validity of the questionnaire.

| Constructs | Items | Mean (SD) | Cronbach's $\alpha$ | Factor Loading | CR | AVE | $\sqrt{\text{AVE}}$ |
|---|---|---|---|---|---|---|---|
| Demographics | 6 | | NA | NA | NA | NA | NA |
| Presence | | | | | | | |
| TP | 9 | 3.96 (0.74) | 0.971 | 0.814–0.930 | 0.970 | 0.785 | 0.886 |
| SP | 7 | 3.87 (0.75) | 0.950 | 0.806–0.893 | 0.950 | 0.733 | 0.856 |
| CP | 8 | 3.89 (0.76) | 0.974 | 0.879–0.933 | 0.974 | 0.825 | 0.908 |
| SRL | | | | | | | |
| Task specific | | | | | | | |
| GS | 3 | 3.76 (0.82) | 0.924 | 0.877–0.914 | 0.926 | 0.805 | 0.897 |
| TS | 3 | 3.76 (0.81) | 0.901 | 0.825–0.889 | 0.903 | 0.757 | 0.87 |
| TM | 3 | 3.80 (0.80) | 0.923 | 0.890–0.903 | 0.923 | 0.801 | 0.895 |
| General | | | | | | | |
| HS | 2 | 3.86 (0.78) | 0.866 | 0.869–0.881 | 0.867 | 0.766 | 0.875 |
| SEV | 2 | 3.81 (0.81) | 0.921 | 0.922–0.926 | 0.921 | 0.854 | 0.924 |
| LM | | | | | | | |
| Interest | 2 | 3.70 (0.87) | 0.645 | 0.879–0.891 | 0.879 | 0.783 | 0.885 |
| Competence | 4 | 3.69 (0.79) | 0.917 | 0.800–0.891 | 0.919 | 0.738 | 0.859 |
| Value | 3 | 3.81 (0.76) | 0.897 | 0.852–0.878 | 0.900 | 0.749 | 0.865 |
| Pressure | 2 | 3.73 (0.84) | 0.713 | 0.822–0.867 | 0.833 | 0.713 | 0.845 |

Note: a. NA means not applicable, demographic variables are not applicable for the reliability and validity analysis; b. TP, teaching presence; SP, social presence; CP, cognitive presence; GS, goal setting; TS, task strategy; TM, time management; HS, help seeking; SEV, self-evaluation; SE, self-efficacy; LM, learning motivation.

### 4.3. Structural Model

The interplay among demographic variables, SRL (task specific SRL, general SRL), CoI presences (TP, SP, and CP), and their impact on LM is demonstrated in Figure 2, with significant SEM results inserted. Furthermore, we examined the model fit between the hypothesized model and empirical data. As shown in Table 4, all the indices indicated good fit: CFI = 0.929, IFI = 0.929 and RMSEA = 0.062, the Chi-square test ($p < 0.001$). We also adopted the standardized root mean square residual (SRMR) to cautiously assess the model fit, and the value of SRMR was lower than 0.08, which was acceptable according to Hair et al. (2017) [83]. The structural relationships of our hypothesized model were well-supported by the empirical data.

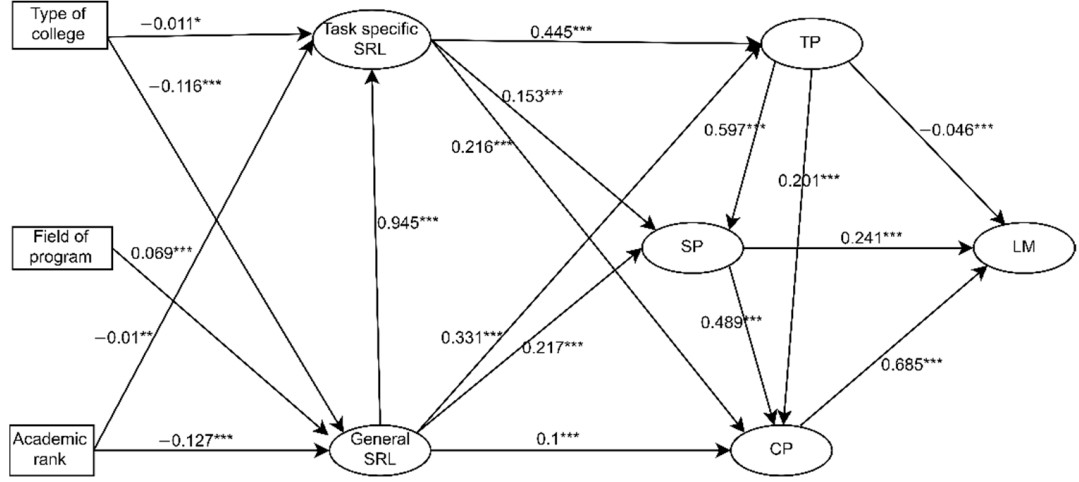

**Figure 2.** Structural model of the relationships among demographic variables, SRL, CoI presences, and LM. * $p < 0.05$; ** $p < 0.01$; *** $p < 0.001$; SRL, self-regulated learning; TP, teaching presence; SP, social presence; CP, cognitive presence; LM, learning motivation.

**Table 4.** Results of the fitness in the hypothesis model.

|  | *p* | CFI | IFI | RMSEA | SRMR |
|---|---|---|---|---|---|
| Structural Model | 0.000 | 0.929 | 0.929 | 0.062 | 0.06 |
| Fit Criteria | <0.001 | >0.9 | >0.9 | <0.08 | <0.08 |

Based on the SEM results, we revised the hypothesized research model by removing non-significant paths between variables and inserted all statistically significant path coefficients in the model. As shown in Figure 2, the majority of the hypotheses were accepted except for a few non-significant paths (e.g., H5a, H5b, and part of H5c). In addition, Table 5 demonstrates the standardized path coefficients, and the direct and indirect effects in the hypothesized model.

**Table 5.** Path coefficient estimates in the hypothesized model.

| Hypothesis | Path Coefficient (β) | Direct Effects | Indirect Effects | SE |
|---|---|---|---|---|
| H1a: TP→SP | 0.597 *** | 0.597 | - | 0.009 |
| H1b: TP→CP | 0.201 *** | 0.201 | - | 0.009 |
| H1c: SP→CP | 0.489 *** | 0.489 | 0.292 | 0.011 |
| H2: General SRL→Task-specific SRL | 0.945 *** | 0.945 | - | 0.007 |
| H3a: TP→LM | −0.046 *** | −0.046 | 0.482 | 0.013 |
| H3b: SP→LM | 0.241 *** | 0.241 | 0.335 | 0.019 |
| H3c: CP→LM | 0.685 *** | 0.685 | - | 0.017 |
| H4a: General SRL→TP | 0.331 *** | 0.331 | 0.42 | 0.024 |
| General SRL→SP | 0.217 *** | 0.217 | 0.593 | 0.017 |
| General SRL→CP | 0.1 *** | 0.1 | 0.751 | 0.014 |
| H4b: Task specific SRL→TP | 0.445 *** | 0.445 | - | 0.025 |
| Task specific SRL→SP | 0.153 *** | 0.153 | 0.266 | 0.018 |
| Task specific SRL→CP | 0.216 *** | 0.216 | 0.294 | 0.015 |
| H5b: Academic rank→General SRL | −0.127 *** | −0.127 | - | 0.007 |
| Academic rank→Task specific SRL | −0.01 ** | −0.01 | −0.12 | 0.003 |
| H5c: Field of program→General SRL | 0.069 *** | 0.069 | - | 0.004 |
| Field of program→Task specific SRL | 0.003 | 0.003 | 0.065 | 0.002 |
| H5e: Type of college→General SRL | −0.116 *** | −0.116 | - | 0.007 |
| Type of college→Task specific SRL | −0.011 * | −0.011 | −0.109 | 0.003 |

Note: *** $p < 0.001$; ** $p < 0.01$; * $p < 0.05$. TP, teaching presence; SP, social Presence; CP, cognitive presence; General SRL, general self-regulated learning; Task-specific SRL, Task-specific self-regulated learning; LM, learning motivation.

### 4.3.1. Hypothesis 1 (Supported): Correlation among the Three CoI Presences

The estimates among three presences indicated that H1a, H1b, and H1c were supported. Teaching presence had the most significant positive effects on social presence (β = 0.597, $p < 0.001$), following by teaching presence on cognitive presence (β = 0.522, $p < 0.001$). In addition, social presence showed significant positive effects on cognitive presence (β = 0.214, $p < 0.001$). Additionally, indirect effects between teaching presence and cognitive presence through social presence were observed (β = 0.292, $p = 0.001$).

### 4.3.2. Hypothesis 2 (Supported): General Self-Regulated Learning Predicts on Task-Specific Self-Regulated Learning

Hypothesis 2 was supported. General self-regulated learning was a significant antecedent to task-specific self-regulated learning (β = 0.945, $p < 0.001$).

### 4.3.3. Hypothesis 3 (Partially Supported): CoI Presences Predict Online Learning Motivation

With regard to the relationships between CoI-related variables and learning motivation, social presence ($\beta = 0.241$, $p < 0.001$) and cognitive presence ($\beta = 0.685$, $p < 0.001$) showed significant positive effects on learning motivation, whereas teaching presence showed significant negative effects on learning motivation ($\beta = -0.405$, $p < 0.001$). Thus, H3b and H3c were supported, H3a was not supported. Inaddition, the indirect effects of teaching presence and social presence on learning motivation were observed.

### 4.3.4. Hypothesis 4 (Supported): Online Students' Self-Regulated Learning Predicts on Their CoI-Presences

Hypothesis 4, that general self-regulated learning and task-specific self-regulated learning positively predicts CoI-presences, was supported. General self-regulated learning had significant positive effects on teaching presence ($\beta = -0.331$, $p < 0.001$), social presence ($\beta = -0.217$, $p < 0.001$), and cognitive presence ($\beta = 0.1$, $p < 0.001$), respectively. Thus, H4a was accepted. In addition, task-specific self-regulated learning had significant positive effects on teaching presence ($\beta = 0.445$, $p < 0.001$), social presence ($\beta = 0.153$, $p < 0.001$), and cognitive presence ($\beta = 0.216$, $p < 0.001$), respectively. Thus, H4b was supported.

### 4.3.5. Hypothesis 5 (Partially Supported): Demographic Factors Predict on Online Students' Self-Regulated Learning

Hypothesis 5 was partially supported. Three demographic factors showed weak effects on self-regulated learning. The academic rank had a significant negative effect on general self-regulated learning ($\beta = -0.127$, $p < 0.001$) and task-specific self-regulated learning ($\beta = -0.1$, $p < 0.001$), thus H5b was not supported. The field of program showed a significant positive effect on general self-regulated learning, and a non-significant positive effect on task-specific self-regulated learning. Thus, H5c was partially accepted. Type of college showed significant negative effects on general self-regulated learning ($\beta = -0.116$, $p < 0.001$) and task-specific self-regulated learning ($\beta = -0.011$, $p < 0.05$). Lastly, sex and seniority appeared to be ineffective predictors of college students' self-regulated learning in online contexts.

## 5. Discussion

### 5.1. Two Types of SRL: Disparity, Similarity, and Influencing Factors

Different from previous studies, this study rejects the unidimensionality of SRL in an online learning context and divides SRL into two related sub-constructs: task-specific SRL and general SRL. While the two sub-constructs were strongly correlated, their effects on cognitive and social aspects of online learning vary. Task-specific SRL has a greater impact on students' cognitive presence, while general SRL has a greater impact on social presence. One possible explanation is that task-specific SRL such as goal setting and task strategies determines the level of student engagement in knowledge construction and higher-order thinking [73], which in turn stimulates the sense of cognitive presence for online learning. Likewise, general SRL promotes social interaction and group cohesion through the communal behaviors of help-seeking [84], and thus leads to enhanced social presence for online students. Despite such a difference, both task-specific and general SRL tend to have the greatest impact on teaching presence, suggesting that students with stronger SRL skills are more likely to discern cues of instructional design and scaffold [85], even in online courses.

Additionally, the present study suggests that students' prior academic experiences cannot effectively predict their online learning SRL, as indicated by the small and insignificant path coefficients. Such a finding is likely due to the fact that most students in this study participated in online learning for the first time, therefore their previously acquired learning experiences and SRL skills might not transfer automatically to the online settings [86]. Another possible reason is that self-reported survey data might not accurately reflect one's SRL

skills, which conceals the potential correlations between prior academic experiences and online SRL. For example, Wu et al. (2019) found that high-achieving Chinese students tend to impose excessive self-demands, leading to low self-evaluation and inhibition behaviors during learning process [87]. This might explain why college rank and academic rank have weak negative effects on SRL in this study.

### 5.2. Varying Impact of CoI Presences on Online Learning Motivation

Contrary to the previous literature that emphasize the importance of teaching presence and social presence for enhanced learning motivation [15,88,89], this study reveals that college students' online learning motivation is largely determined by their cognitive presence during the learning process ($\beta$ = 0.685). Such a finding also contradicts a similar study conducted by Zuo et al. (2021a) among K-12 students, in which the effect of cognitive presence on online learning motivation was much smaller ($\beta$ = 0.176) [90]. One possible explanation is that, compared to younger students, adult learners have a greater disposition to engage in higher-level cognitive activities during online learning [91,92] and value intellectual challenge as a source of intrinsic motivation [93,94].

The fact that social presence has only a moderate effect on online learning motivation is not surprising, and there are cultural and technical reasons behind such a finding. Compared to the Western world, education culture in China is more goal-oriented and outcome-driven [95], where the process of social interaction and collaboration has not been emphasized historically. In fact, many Chinese students prefer individual learning over group work for concerns of efficiency and fairness [96,97]. Additionally, the sudden transition to online education also posed technical challenges for sustaining social learning: many online learning platforms were found to lack proper functions to support collaborative inquiry [5,90]. Consequently, the depreciation of social presence has led to its diminishing influence on students' online learning motivation.

Interestingly, teaching presence turns out to have the weakest effect ($\beta$ = −0.046) on college students' online learning motivation. Such a discovery is consistent with Zuo et al.'s (2021a) finding with K-12 students ($\beta$ = −0.038) [90], indicating the ubiquity of this phenomenon. One possible reason is that online teaching presence lacks variation due to the homogeneous instructional design afforded by the highly standardized structure and layout of online learning platforms [5,98]. The pervasively high rating of teaching presence actually makes it a poor predictor of motivation. However, it is important to note that teaching presence has a positive indirect effect on online learning motivation ($\beta$ = 0.482), proving that quality instructional design can change students' opinion towards online learning through well-designed social and cognitive events [99].

### 5.3. A Key Pathway of Influence

This study shows that college students' SRL skills can influence their online learning motivation through perceived cognitive and social presence in online courses. Among the various pathways from SRL to motivation, the path of "task-specific SRL → CP → LM" deserves special attention as it generates the largest positive compound effect. While the close relationship among SRL, cognitive learning, and motivational beliefs was emphasized by Pintrich (1999), this key pathway reveals the reversed role of SRL in sustaining online learning motivation and highlights the importance of contextually bounded SRL skills such as goal setting, task strategies, and time management [11]. Compared to habitual dispositions, those SRL skills are essential to direct and regulate students' cognitive processes during online learning [35]. Moreover, the mediating role of cognitive presence indicates that the key to maintain college students' motivation in online courses is to convert their time and efforts in self-regulation into cognitive learning opportunities. Such a finding agrees with previous literature that recognizes the importance of cognitive engagement in online learning environments, as it can significantly enhance students' satisfaction and course retention [100,101], which are considered critical indicators of learning motivation.

## 6. Conclusions and Implications

In conclusion, this study proposed and validated a conceptual model for predicting college students' online learning motivation based on a national survey during COVID-19, with SRL and first-time online learning experiences being the key predictors. The study results indicate that students' SRL in an online learning context cannot be effectively predicted by their prior academic achievement, as measured by college type and academic rank. It can be further differentiated into general and task-specific strategies, which have a varying impact on three CoI presences. The study results also highlight the importance of cognitive presence, as it has the largest effect on online learning motivation. Contrarily, teaching presence proves to have insubstantial predicting power despite its high ratings. Lastly, the path of task-specific self-regulated learning → cognitive presence → online learning motivation has been identified, as it generates the largest positive compound effect.

Based on the research findings, we propose the following implications for college students, teachers, and platform developers to further improve the online teaching and learning practice in a higher education context. For college students, they should be aware of the challenges of online learning and importance of self-regulation. The self-directedness and student-centered nature of online learning emphasize the importance of SRL, and students' SRL behaviors need to be adjusted according to different learning tasks. For teachers, they should not assume that high-achieving students possess better SRL skills. As a result, proper guidance on SRL strategies (e.g., goal setting and help seeking netiquette) should be offered to students in the early phase of online instruction. Additionally, teachers should find creative approaches to incorporate more higher-order thinking opportunities in online learning processes, due to the importance of cognitive presence for online learning motivation. For platform developers, online learning platforms should afford sufficient flexibility and customization for teachers to enable more individualized instructional design. Moreover, key functions for social learning and collaboration (e.g., thumbs up, social network integration, and convenient synchronous and asynchronous communication) should be included in online learning platforms to boost both social and cognitive presence.

## 7. Limitations and Future Research

There are several limitations of the current study that should be addressed when interpreting the research results. First, the data source of the survey were self-reported perceptions based on students' one semester of online learning. Thus, it may be questionable about the data validity and run a risk response bias [102]. Accordingly, we recommend future studies to include multiple sources of data, such as students' behavioral data (e.g., learning analytics for SRL and motivation), students' interviews, and online test scores. The second limitation is the representativeness of the survey sample. Although the sample size is relatively large, it is biased to a few colleges located in central region of China, and thus may not be generalized with more diverse student populations. Therefore, we recommend using stratified sampling in future research to boost the representativeness of the survey sample and the generalizability of the research findings. Third, the dataset we analyzed is one-time and quantitative only. More supplementary qualitative data in future studies should be used to assist meaningful interpretation of the statistical survey results. Moreover, more longitudinal studies are needed to examine how students' online learning motivation and the influencing factors might change over time.

**Author Contributions:** Conceptualization, H.L. and G.L.; methodology, G.L. and J.L.; formal analysis, G.L., S.X. and T.C.; investigation, G.L. and H.L.; writing—original draft preparation, G.L., H.L. and S.X.; writing—review and editing, J.L. and T.C.; supervision, H.L. and J.L.; project administration, H.L.; funding acquisition, H.L. All authors have read and agreed to the published version of the manuscript.

**Funding:** Please add: This research was funded by Teacher Education Specialized Grant of Central China Normal University, grant number CCNUTEIII 2021-10.

**Institutional Review Board Statement:** Ethical review and approval were waived for this study because it is a non-interventional study based solely on survey data. IRB approval was exempted by the Institutional Review Board of Central China Normal University on 2020/07/02.

**Informed Consent Statement:** Informed consent was obtained from all subjects involved in the study. All participants were fully informed if the anonymity would be assured, why the research was being conducted, how their data would be used.

**Data Availability Statement:** The data presented in this study are openly available in Mendeley Data at https://doi.org/10.17632/zzc894hydt.1.

**Acknowledgments:** We would like to thank all students who voluntarily participated in our survey research by submitting their questionnaire responses during COVID-19 pandemic.

**Conflicts of Interest:** The authors declare no conflict of interest.

## Appendix A. Questionnaire

Introduction: Greetings! We would like to invite you to participate in our survey on college students' online learning experiences during the suspension of 2020 Spring Semester. Please answer the following survey questions truthfully based on your online learning experience. The information we collect from this survey will be used for research purpose only, and any personally identifiable information will be removed from all publications and presentations. Your participation in the survey is voluntary. Thank you for your participation!

*Basic information*

1. Your birth sex is
   ○ Male ○Female
2. Your age is ( )
3. Which seniority are you currently in?
   ○Year 1 undergraduate ○Year 2 undergraduate ○Year 3 undergraduate
   ○Year 4+ undergraduate ○Graduate
4. Which of the following descriptions best applies to the college you are currently attending?
   ○Top tier ○Second tier ○Third tier ○Vocational college ○Other
5. Your field of program is?
   ○Philosophy ○Economics ○Law ○pedagogics ○Social science
   ○History ○Science ○Engineering ○Agronomy ○Medical science
   ○Management ○Arts ○Military ○Other
6. Your current academic rank is?
   ○Top rank ○Above average ○Medium ○Below average ○Ranking low

*Self-regulated learning (SRL)*

Goal setting

1. I set short-term (daily or weekly) goals as well as long-term (monthly or termly) goals for online learning.
2. I set high standards for my learning when studying online.
3. I do not compromise the quality of my work because it is online.

Task strategy

1. I take more thorough notes during online learning.
2. I often participated in online discussions in class with questions.

3. Except the assigned content, I work extra questions or readings to master the course content.

Time management

1. I allocate extra studying time for my online courses because I know it is time-demanding.
2. I try to schedule the time every day or every week to study for my online courses.
3. Although we do not have to attend daily classes, I still try to distribute my studying time evenly across days.

Help seeking

1. I will take the initiative to contact the course teacher for answers so that I can consult with him or her when I need help.
2. I share my problems with my classmates online so we can solve problems together.

Self-evaluation

1. I ask myself a lot of questions about the course material when studying online.
2. I summarize my learning in online courses regularly to examine my learning effectiveness.

*CoI-related presences*
Teaching presence

1. The instructor clearly communicated important course goals.
2. The instructor provided clear instructions on how to participate in course learning activities.
3. The instructor clearly communicated important due dates/time frames for learning activities.
4. The instructor helped to keep course participants engaged and participating in productive discussion.
5. The instructor helped keep the course participants on task in a way that helped me to learn.
6. Instructor actions reinforced the development of mutual help and recognition among course participants.
7. The instructor helped to focus discussion on relevant content in a way that helped me to learn.
8. The instructor provided feedback that helped me understand my strengths and weaknesses relative to the online learning of the courses.
9. The instructor provided feedback in a timely fashion.

Social presence

1. I was able to form distinct impressions of some online course participants.
2. Online or web-based communication is an excellent medium for social interaction.
3. I felt comfortable conversing through the online medium.
4. I felt comfortable participating in the online course discussions.
5. I felt comfortable disagreeing with other online course participants while still maintaining a sense of trust.
6. I felt that my point of view was acknowledged by other online course participants.
7. Online discussions help me to develop a sense of collaboration.

Cognitive presence

1. Learning tasks in online courses increased my interest in learning.
2. Course activities piqued my curiosity.
3. I felt motivated to explore content related questions.
4. I was able to explore and complete the learning tasks in the course using a variety of materials.
5. Online discussions were valuable in helping me appreciate different perspectives.
6. Learning activities of courses helped me construct knowledge and problem solutions.
7. Reflection on course content and discussions helped me understand fundamental concepts of this course.
8. I can apply the knowledge created in this course to my work or other non-class related activities.

*Learning motivation*

Interest

1. I think online learning tasks are very interesting.
2. I think online learning activities are boring.
3. I enjoyed the online learning content very much.

Competence

1. I am competent for the online learning tasks.
2. I think I did pretty well at online activities, compared to other participants.
3. I am satisfied with my performance at online learning task.
4. I performed well during the learning of online course.

Value

1. I believe online learning activities could be of some value to me.
2. I think that conducting online learning activities helps to better understand my profession.
3. I believe online learning activities could be beneficial to me.

Pressure

1. I did not feel nervous at all during online learning.
2. I felt very relaxed in learning the content of online courses.
3. I felt very anxious when performing online learning tasks.

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
