# Peer review of "Effects of First-Time Experiences and Self-Regulation on College Students’ Online Learning Motivation: Based on a National Survey during COVID-19"

_education, doi:10.3390/educsci12040245_

Round 1

Reviewer 1 Report

A very interesting and well-executed manucript, with a highly topical subject.

In terms of format, the numbering of the sections should be revised, especially on page 7, where there is confusion in the numbering. There are also some typographical errors. 

The theoretical framework includes a broad theoretical approach with well-selected references. 

However, I recommend the authors to remove the hypotheses from the theoretical framework and move them to the Method section, where they should be located.

The method section is well written but lacks information on the validation of the instruments used. This information should be added. It is appreciated that they have included the instrument in the annexes.

Although it is a very complete section, I recommend separating discussion and conclusion as two sections.

I congratulate the authors for the manuscript and hope to contribute to its improvement with these recommendations. 

Reviewer 2 Report

Dear author(s),

congratulation on you impressive collection of data. I was not sure you comply with the academic rigours of analyzing the data like validity and representativity, but the last chapter "saved" from rejection.

What is the added value of this study compared to similar studies? Originality of the study?

Line 32-33- the author(s) state that:

"the success of online learning is largely determined by three pertinent constructs: learner motivation, educational experience, and self-regulation [6-8]. what about barriers? lack of internet access? No money for laptops or computers?

Further good luck and all the best
